Subject Areas:
civil engineering/engineering geology/power and energy systems

Keywords:
hard rock, fracturing behaviour, extra-thick coal seam, impact characteristics

Authors for correspondence:
Rui Gao
e-mail: cumtgaorui@163.com
Bingjie Huo
e-mail: huobingjie@163.com

# Numerical simulation on fracturing behaviour of hard roofs at different levels during extra-thick coal seam mining

Rui Gao[1], Bingjie Huo[2], Hongchun Xia[3] and Xiangbin Meng[4]

[1]College of Mining Engineering, Taiyuan University of Technology, Shanxi 030024, People's Republic of China
[2]College of Mining Engineering, Liaoning Technical University, Fuxin 123000, People's Republic of China
[3]College of Civil and Architectural Engineering, Dalian University, Dalian 116622, People's Republic of China
[4]Datong Coal Mine Group Co. Ltd., Datong 037000, People's Republic of China

RG, 0000-0003-1988-8778

In fully mechanized caving mining of extra-thick coal seams, the movement range of overburden is wide, resulting in the breakage of multilayer hard roofs in overlying large spaces. However, the characteristics, morphology and impact effect of hard roofs at different levels are different and unclear. In this study, a secondary development was used in the numerical simulation software ABAQUS, and the caving of rock strata in the finite-element software was realized. The bearing stress distribution, fracturing morphology and impact energy characteristics of hard roofs at different levels were studied to reflect the action and difference of hard roof failure on the working face; thus, revealing the mechanism of the strong ground pressure in stopes, and providing a theoretical basis for the safe and efficient mining of extra-thick coal seams with hard roofs.

## 1. Introduction

Thick and extra-thick coal seams are the primary coal seams of high-efficiency mining, for which the caving mining method is primarily used. Owing to the large thickness of coal seams, the range of overburden migration is wide. Previous studies have shown that the zone of overburden failure could be greater than 300 m during a 20 m thick coal seam mining [1,2]. When multiple hard roofs exist in the overburden, the fracturing and

breaking down of the hard roofs at different levels will result in the frequent occurrence of a strong ground pressure in stopes, such as crashed supports and damaged roadways that significantly affects the safety of production [3,4]. Therefore, to ensure the safe and efficient production of an extra-thick coal seam and propose specific technical measures to prevent strong ground pressures, it is necessary to study the pressure mechanism induced by the breaking of multiple hard roofs. Research has been conducted in related areas that consider the effect of strong ground pressure in extra-thick coal seam mining with hard roofs. Ning et al. [5] analysed the fracturing characteristics of overlying hard roofs by microseismic monitoring that explained the strong strata behaviour induced by thick and hard roofs. According to Zhu's [6] study, the rotary movement of the key strata (KS) in an overburden directly affected the pressure appearance of the supports in the working face, and the effect of the rotary angle of the KS at different levels on the stope pressure was analysed. In addition, the support crashing mechanism was studied during the mining of an extra-thick coal seam with hard roofs, indicating that the high hardness of the top coal and the insufficient resistance of the supports were the primary causes of the crashed supports [7]. Li et al. [8] adopted the thick-plate theory to study the fracturing characteristics and the span of thick and hard roofs, and microseismic monitoring was used to analyse the energy characteristics of roof fracturing. Zhenlei et al. [9] compared the occurrence of rockburst in stopes when the mining methods of top-coal caving and slicing mining were used. Xie et al. [10] analysed the effect of different thicknesses and distributions of hard roofs on the peak value and influence range of leading abutment stress of the working face. Similar to the above research, studies on the effect of hard roof fracturing on stope pressure began from a local perspective. For the mining of an extra-thick coal seam, the hard roofs at different levels would break owing to the large mining thickness. According to the KS theory [11,12], the breaking span and distance from the coal seam vary with the level of the KS. The higher the KS level, the greater the breaking span. However, current studies have not elaborated the impact strength of KS at different levels; therefore, further research is required.

The method of numerical simulation can directly reflect the structural characteristics of strata fracturing, strength of ground pressure, and stress distribution [10,13–17]. Currently, the UDEC software is widely used to simulate rock fracturing. However, when UDEC is used to perform simulations, an artificial block division of the strata should be performed in advance. Block division determines the fracture step of the strata; therefore, the actual fracture situation of the strata is not well reflected [14–17]. In addition, the finite-element analysis software FLAC is primarily used to study the distribution of the abutment stress in a stope, which is not ideal for simulating the fracturing effect of rock strata [18–22]. Therefore, a good research method to reflect the fracture characteristics of rock strata and the corresponding action of ground pressure is necessary.

In this study, the numerical simulation software ABAQUS was adopted to reflect the fracture law and structural characteristics of rock strata by combining the advantages of finite and discrete element analyses [23] and setting reasonable parameters of the rock strata. Herein, the typical hard roof mining area in Datong, China is used as a study example. The coal seam thickness is up to 20 m, and the simulation study is primarily aimed at the fracture characteristics of hard roofs at different levels and the corresponding effects of pressure, to reveal the mechanism of strong ground pressures on the working face and provide a theoretical basis for the control of hard roofs.

# 2. Numerical simulation model

## 2.1. Background

The no. 3–5 extra-thick coal seam is mainly mined in the Tashan coal mine of the Datong mining area; the coal seam thickness ranges from 14 to 20 m and the burial depth ranges from 400 to 800 m. The overlying rock comprises multiple hard roofs with compressive strengths of 60–120 MPa. Considering the 8102 working face in the Tashan coal mine as an example, the mined coal seam thickness, burial depth and dip are 20 m, 470 m and 1–3°, respectively. The working face is 230 m long and the continuous mining length is 1500 m. The working face is covered with multiple hard roofs, as shown in figure 1.

## 2.2. Numerical modelling

A two-dimensional numerical model of size 492 m × 600 m was established and was built from the coal seam to the ground surface. To eliminate the effect of the boundary, a 50 m coal pillar was set, and the

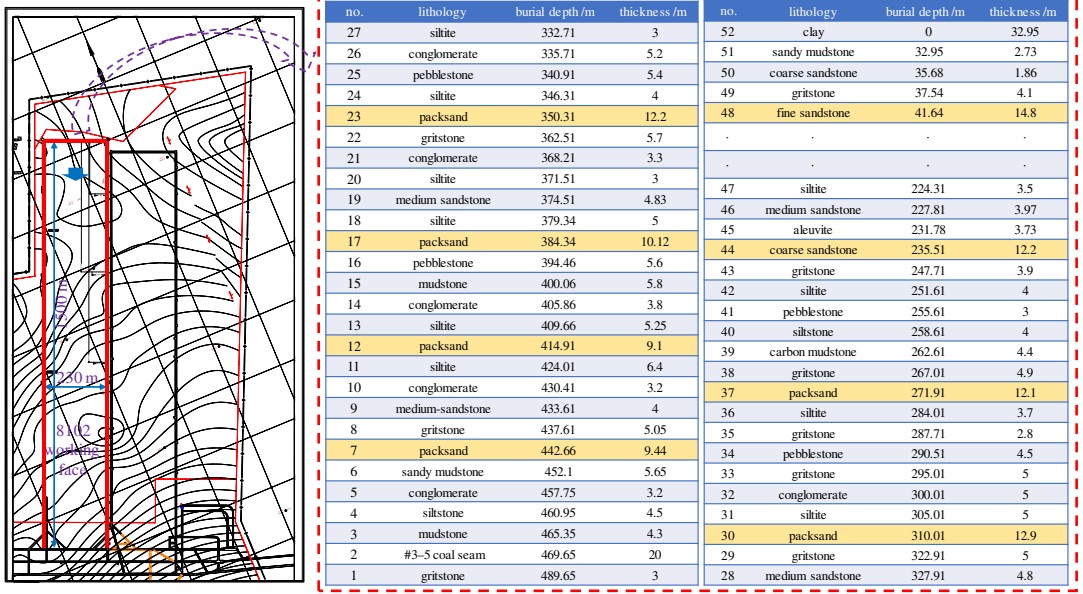

| no. | lithology | burial depth /m | thickness /m | no. | lithology | burial depth /m | thickness /m |
|---|---|---|---|---|---|---|---|
| 27 | siltite | 332.71 | 3 | 52 | clay | 0 | 32.95 |
| 26 | conglomerate | 335.71 | 5.2 | 51 | sandy mudstone | 32.95 | 2.73 |
| 25 | pebblestone | 340.91 | 5.4 | 50 | coarse sandstone | 35.68 | 1.86 |
| 24 | siltite | 346.31 | 4 | 49 | gritstone | 37.54 | 4.1 |
| 23 | packsand | 350.31 | 12.2 | 48 | fine sandstone | 41.64 | 14.8 |
| 22 | gritstone | 362.51 | 5.7 | . | . | . | . |
| 21 | conglomerate | 368.21 | 3.3 | | | | |
| 20 | siltite | 371.51 | 3 | . | . | . | . |
| 19 | medium sandstone | 374.51 | 4.83 | 47 | siltite | 224.31 | 3.5 |
| 18 | siltite | 379.34 | 5 | 46 | medium sandstone | 227.81 | 3.97 |
| 17 | packsand | 384.34 | 10.12 | 45 | aleurite | 231.78 | 3.73 |
| 16 | pebblestone | 394.46 | 5.6 | 44 | coarse sandstone | 235.51 | 12.2 |
| 15 | mudstone | 400.06 | 5.8 | 43 | gritstone | 247.71 | 3.9 |
| 14 | conglomerate | 405.86 | 3.8 | 42 | siltite | 251.61 | 4 |
| 13 | siltite | 409.66 | 5.25 | 41 | pebblestone | 255.61 | 3 |
| 12 | packsand | 414.91 | 9.1 | 40 | siltstone | 258.61 | 4 |
| 11 | siltite | 424.01 | 6.4 | 39 | carbon mudstone | 262.61 | 4.4 |
| 10 | conglomerate | 430.41 | 3.2 | 38 | gritstone | 267.01 | 4.9 |
| 9 | medium-sandstone | 433.61 | 4 | 37 | packsand | 271.91 | 12.1 |
| 8 | gritstone | 437.61 | 5.05 | 36 | siltite | 284.01 | 3.7 |
| 7 | packsand | 442.66 | 9.44 | 35 | gritstone | 287.71 | 2.8 |
| 6 | sandy mudstone | 452.1 | 5.65 | 34 | pebblestone | 290.51 | 4.5 |
| 5 | conglomerate | 457.75 | 3.2 | 33 | gritstone | 295.01 | 5 |
| 4 | siltstone | 460.95 | 4.5 | 32 | conglomerate | 300.01 | 5 |
| 3 | mudstone | 465.35 | 4.3 | 31 | siltite | 305.01 | 5 |
| 2 | #3–5 coal seam | 469.65 | 20 | 30 | packsand | 310.01 | 12.9 |
| 1 | gritstone | 489.65 | 3 | 29 | gritstone | 322.91 | 5 |
| | | | | 28 | medium sandstone | 327.91 | 4.8 |

**Figure 1.** Occurrence condition of working face and overburden.

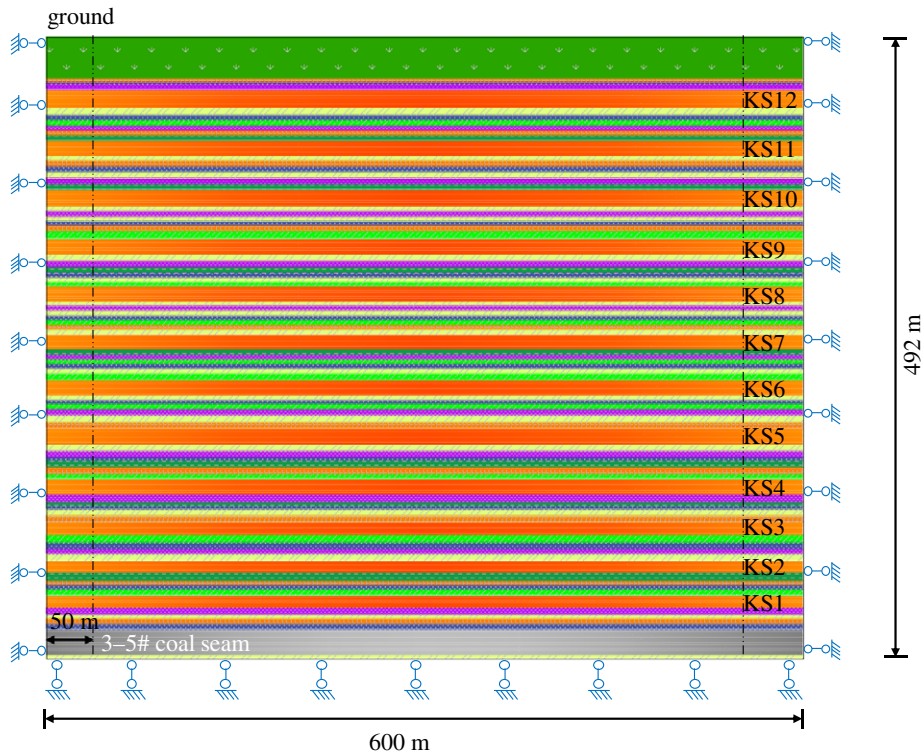

**Figure 2.** Numerical simulation model.

working face was mined continuously for a length of 500 m. The mining thickness of the coal seam was 20 m, and the mesh size was between 3 m and 6 m, with a total of 14 000 grids. The horizontal displacements on both sides of the model and the horizontal and vertical displacements at the bottom were fixed, as shown in figure 2.

The basic parameters of each layer were obtained by a field drilling and coring test; because of the existing joints and cracks in the panel, the measured parameters in the laboratory are generally higher than those of the *in situ* rock mass. Thus, the rock quality designation (RQD) index was introduced to obtain the mechanical parameters of the *in situ* rock mass and was evaluated by field drilling.

The relationship between the elastic modulus $E_m$ of the intact rock in the laboratory and the elastic modulus $E_r$ of the rock mass is defined as follows [24]:

$$\frac{E_m}{E_r} = 10^{0.0186RQD-1.91}. \tag{2.1}$$

The ratio of the compressive strength $\sigma_{cm}$ of the laboratory intact rock to the compressive strength $\sigma_c$ of the field rock mass has the following relationship with $E_m/E_r$:

$$\frac{\sigma_{cm}}{\sigma_c} = \left(\frac{E_m}{E_r}\right)^q, \tag{2.2}$$

where $q$ is an empirical parameter equal to 0.7 in this study.

The mechanical parameters of overlying roofs are shown in table 1.

The horizontal stress measured in the field was 1.2 times the vertical stress by the method of hydraulic fracturing. Meanwhile, according to Kang's study [25], the average horizontal stress and vertical stress obey the following relationship:

$$K_{av} = \frac{129.58}{H} + 0.606, \tag{2.3}$$

where $K_{av}$ is the ratio of the horizontal stress to the vertical stress, and $H$ is the burial depth.

To add an average coefficient of lateral pressure in the model, five values of the burial depth were selected as 490, 390, 290, 190 and 90 m, and $K_{av}$ of 1.2 was obtained. Thus, the $K_{av}$ value of 1.2 was added in the simulation, and the lateral pressure coefficient was 1.23 when the initial geostress was generated in the simulation process.

We used the powerful engineering simulation software ABAQUS that provides a cohesive element to simulate a rock's tensile and shear failure behaviours; it comprises two types of failure models: the traction separation criterion model and the continuum criterion model. The traction separation model is widely used because of its ease of operation.

For the tensile properties, the bilinear constitutive model has been most commonly used [26]. As shown in the first phase boundary in figure 3, the cohesive constitutive model of the cohesive unit was determined by defining the rock stiffness, ultimate strength and critical breaking energy release rate. As shown in the second phase boundary of figure 3, the bilinear constitutive model is useful for shear failure, and the maximum nominal stress failure criterion was adopted in this study. The Mohr Coulomb criterion was used in the model.

To simulate the fracturing and caving of the rock strata, a secondary development was conducted in the model. First, the rand function was used to generate the corresponding slices in the model to simulate the cracks in the rock strata. Then a cohesive unit with zero thickness was inserted at each slice. The constitutive model of the cohesive element was Mohr Coulomb, which is shown as figure 3, and a VUSDFLD subroutine was used to invoke the function; thus, realizing the fracturing and caving of the rock strata.

# 3. Result analysis

## 3.1. Distribution of abutment stress

### 3.1.1. Stress distribution in hard roofs during primary critical fracture

As shown in figure 4, the distribution of the abutment stress on the upper surface of each hard roof before its primary critical fracture was obtained. Because this numerical calculation realized the fracturing of the rock strata by inserting joint planes into the finite-element unit, the stress distribution of the rock strata presented certain dispersions and deviations; however, the distribution law was still clear, and could reflect the stress distribution of the rock strata.

Figure 4a shows the distribution characteristics of the bearing stress before the critical fracture of KS1. As shown in the figure, the overlying bearing stress was distributed in an inverted 'V' shape in the overhanging section of the hard roof, with a large vertical stress at its centre, indicating that in the overhanging section of the hard roof, the overlying load is transferred to the centre, and the stress on both sides of the overhanging section was at a minimum value of 0 MPa. As the rock mass moves away gradually from the overhanging section, the overlying bearing stress increases gradually, and it

**Table 1.** Parameters of coal and rock masses.

| lithology | thickness (m) | compressive strength (MPa) | tensile strength (MPa) | elasticity modulus (GPa) | Poisson ratio | internal friction angle (°) | cohesion (MPa) |
|---|---|---|---|---|---|---|---|
| clay | 32.95 | 10.3 | 1.1 | 5.5 | 0.3 | 22 | 2.3 |
| sandy mudstone | 2.73 | 23.71 | 3.3 | 12.5 | 0.26 | 21 | 4.1 |
| coarse sandstone | 1.86 | 34.43 | 3.9 | 28 | 0.24 | 22 | 4.2 |
| gritstone | 4.1 | 24.7 | 3.4 | 19 | 0.27 | 21 | 3.9 |
| fine sandstone | 14.8 | 91.39 | 8.3 | 61 | 0.22 | 24 | 9.1 |
| · | · | · | | | | | |
| · | · | · | | | | | |
| · | · | · | | | | | |
| siltite | 3.5 | 32.9 | 3.1 | 22 | 0.27 | 23 | 4.9 |
| medium sandstone | 3.97 | 38.6 | 3.6 | 19.5 | 0.25 | 24 | 5.1 |
| aleuvite | 3.73 | 35.6 | 3.5 | 18.5 | 0.27 | 23 | 4.8 |
| coarse sandstone | 12.2 | 95.35 | 11.2 | 60 | 0.23 | 25 | 8.2 |
| gritstone | 3.9 | 22.9 | 2.2 | 8 | 0.3 | 22 | 3.5 |
| siltite | 4 | 30.87 | 2.9 | 20 | 0.29 | 22 | 4.8 |
| pebblestone | 3 | 33.32 | 3.1 | 22 | 0.27 | 21 | 5.1 |
| siltstone | 4 | 23.67 | 2.4 | 15 | 0.27 | 21 | 4.3 |
| carbon mudstone | 4.4 | 15.23 | 1.3 | 13 | 0.21 | 22 | 3.7 |
| gritstone | 4.9 | 28.34 | 3.9 | 20 | 0.29 | 23 | 3.2 |
| packsand | 12.1 | 90.53 | 12.3 | 65 | 0.21 | 26 | 8.5 |
| siltite | 3.7 | 33.6 | 4.3 | 26 | 0.23 | 22 | 5.3 |
| gritstone | 2.8 | 23.1 | 3.3 | 16.5 | 0.26 | 21 | 3.9 |
| pebblestone | 4.5 | 35.6 | 3.9 | 28 | 0.24 | 22 | 5.2 |
| gritstone | 5 | 25.9 | 3.4 | 17 | 0.27 | 21 | 4.8 |
| conglomerate | 5 | 37.74 | 4.1 | 19.3 | 0.23 | 22 | 6.2 |
| siltite | 5 | 24.1 | 3.4 | 18 | 0.27 | 21 | 3.9 |
| packsand | 12.9 | 80.21 | 8.2 | 60.5 | 0.22 | 25 | 8.1 |
| gritstone | 5 | 26.6 | 3.4 | 19 | 0.27 | 22 | 3.9 |
| medium sandstone | 4.8 | 30.3 | 2.3 | 17.2 | 0.29 | 21 | 4.8 |
| siltite | 3 | 35.87 | 3.3 | 13 | 0.26 | 23 | 5.2 |
| conglomerate | 5.2 | 25.6 | 3.4 | 19.2 | 0.27 | 22 | 4.3 |
| pebblestone | 5.4 | 32.3 | 2.3 | 27.5 | 0.29 | 23 | 3.7 |
| siltite | 4 | 23.57 | 3.4 | 23 | 0.27 | 22 | 3.2 |
| packsand | 12.2 | 80.27 | 7.8 | 60 | 0.20 | 24 | 8 |
| gritstone | 5.7 | 32.87 | 3.6 | 24.6 | 0.26 | 22 | 5.6 |
| conglomerate | 3.3 | 22.9 | 3.4 | 13 | 0.27 | 21 | 4.8 |
| siltite | 3 | 33.85 | 3.6 | 24.1 | 0.26 | 22 | 5.6 |
| medium sandstone | 4.83 | 30.24 | 2.3 | 17.6 | 0.29 | 23 | 5.2 |
| siltite | 5 | 23.23 | 3.4 | 12.7 | 0.27 | 22 | 4.3 |
| packsand | 10.12 | 65.53 | 6.1 | 45.5 | 0.21 | 26 | 7.6 |
| pebblestone | 5.6 | 30.23 | 2.4 | 19.5 | 0.29 | 20 | 4.6 |

(*Continued.*)

| lithology | thickness (m) | compressive strength (MPa) | tensile strength (MPa) | elasticity modulus (GPa) | Poisson ratio | internal friction angle (°) | cohesion (MPa) |
|---|---|---|---|---|---|---|---|
| mudstone | 5.8 | 25.3 | 2.3 | 13.5 | 0.27 | 21 | 3.8 |
| conglomerate | 3.8 | 29.13 | 2.3 | 17.5 | 0.29 | 22 | 4.9 |
| siltite | 5.25 | 36.56 | 2.8 | 19.8 | 0.28 | 21 | 5.3 |
| packsand | 9.1 | 55 | 4.2 | 40.5 | 0.23 | 25 | 8.0 |
| siltite | 6.4 | 30.73 | 2.4 | 19.3 | 0.29 | 23 | 6.1 |
| conglomerate | 3.2 | 23.6 | 3.4 | 13.1 | 0.27 | 21 | 5.9 |
| medium sandstone | 4 | 30.43 | 2.3 | 17.5 | 0.29 | 22 | 5.4 |
| gritstone | 5.05 | 38.6 | 2.6 | 20.1 | 0.26 | 23 | 5.3 |
| packsand | 9.44 | 55.53 | 5.9 | 28.5 | 0.23 | 25 | 6.2 |
| sandy mudstone | 5.65 | 33.3 | 2.9 | 10.5 | 0.27 | 22 | 5.3 |
| conglomerate | 3.2 | 23.34 | 3.4 | 13.2 | 0.27 | 23 | 5.1 |
| siltstone | 4.5 | 21.07 | 2.5 | 12.6 | 0.32 | 22 | 4.3 |
| mudstone | 4.3 | 17.36 | 2.2 | 8 | 0.32 | 21 | 4.2 |
| no. 3–5 coal seam | 20 | 15.94 | 1.1 | 7.5 | 0.33 | 20 | 4.0 |
| gritstone | 3 | 43.87 | 3.2 | 17 | 0.22 | 22 | 4.1 |

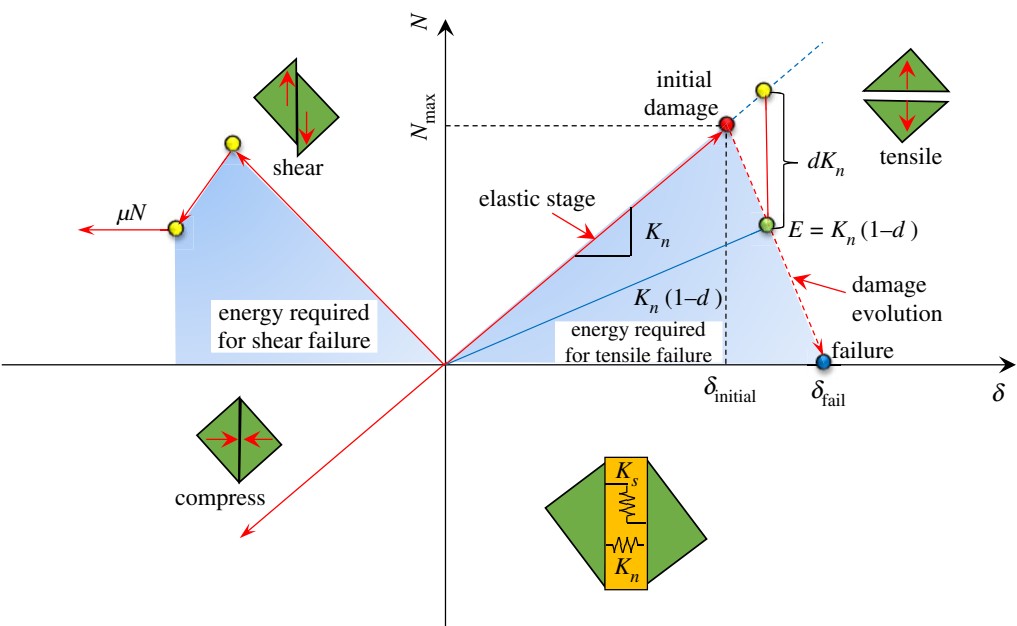

**Figure 3.** Criterion of traction separation.

then decreases before becoming stable. At a distance of 40 m from the side of the overhanging section, the lateral supporting stress reached the peak.

Figure 4*b* shows the distribution characteristics of the bearing stress before the critical fracture of KS2. The stress distribution characteristics in KS2 are similar to those in KS1, with the minimum stress at both ends of the overhanging section and the maximum stress at the middle. At 48 m from one side of the overhanging section, the lateral stress reached the peak. Compared with KS1, the lateral peak stress increased to a certain extent.

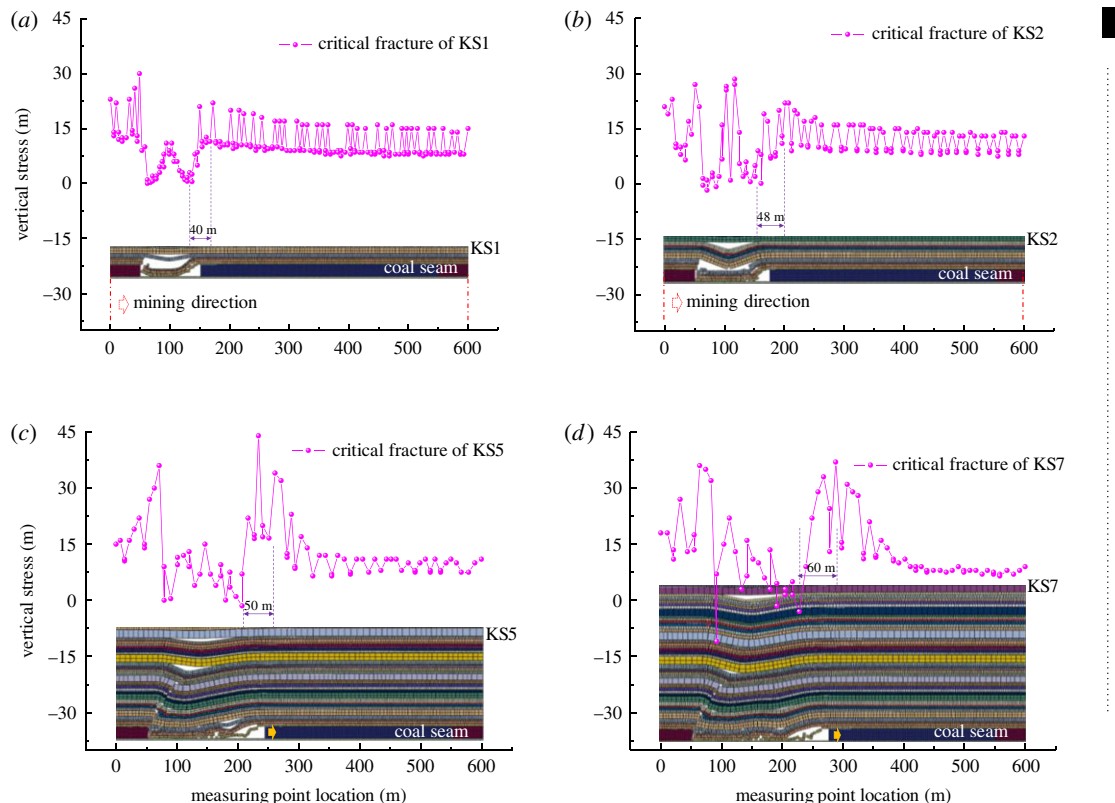

**Figure 4.** Stress distributions in the key strata (KS). Stress distribution in (*a*) KS1, (*b*) KS2, (*c*) KS5 and (*d*) KS7.

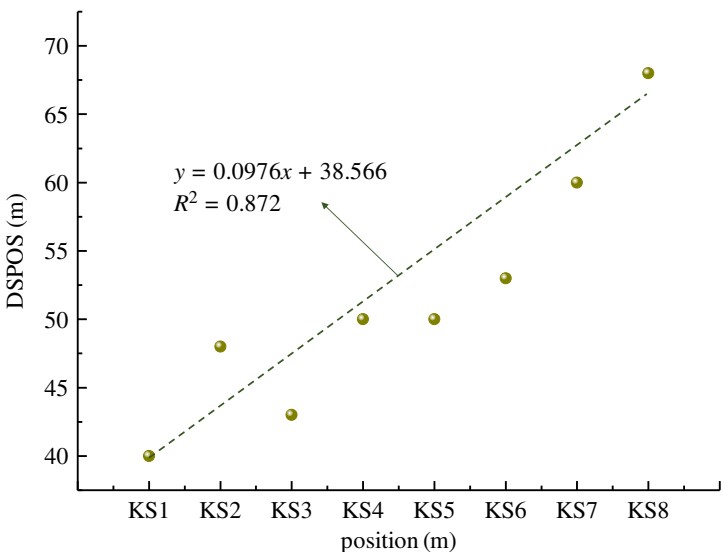

**Figure 5.** Peak range of key strata (KS) varying with distance to coal seam.

As shown in figure 4*c*,*d*, with the continued mining of the working face, the overburden failure developed gradually upwards, and the hard roofs of KS3 to KS8 successively broke. The stress distribution law on each hard roof is similar to those of KS1 and KS2. Through the simulations, it was found that with the increase in the overburden fracturing height, the stress distributed on the hard roofs tends to increase, but no apparent regularity was found. This is primarily because the load on the hard rock is closely related to the roof thickness, hard rock distribution characteristics and overburden thickness. However, the results indicate that the distance between the stress peak and the overhanging section (DSPOS) in each roof increases with the fracturing height. According to statistics, DSPOS in the vertical distance between the hard roof and coal seam is shown in figure 5.

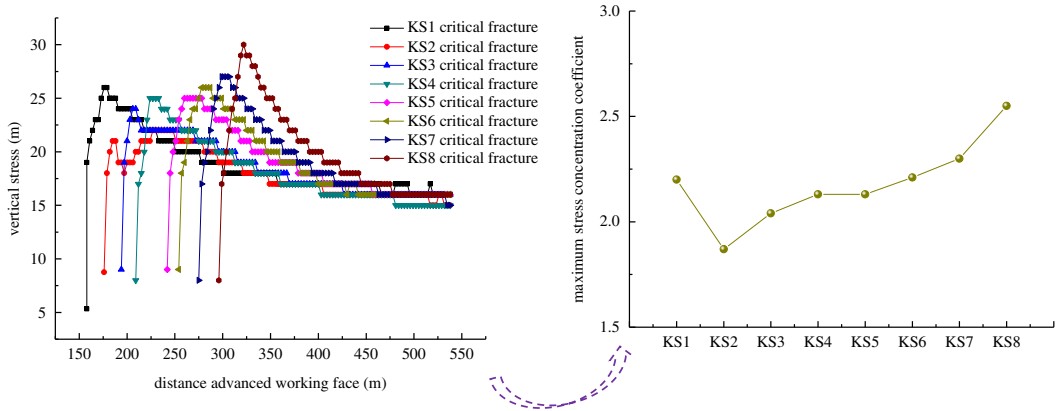

**Figure 6.** Variation in stress and concentration coefficient of the advanced coal body.

As shown in figure 5, DSPOS in KS1 was 40 m, while in KS8 it was 68 m, indicating that the influence range of the bearing stress in the hard roof increases with the fracturing height; however, the effect of the bearing stress on the coal body in front of the working face still requires analysis.

### 3.1.2. Stress distributed in the advanced coal body

According to statistics, the stress distribution on the coal body in front of the working face when the hard roof is critically fractured for the first time is as shown in figure 6. The latter shows that with the increase in the overburden fracture height, the peak stress in the advanced coal body exhibits a gradually increasing trend. The maximum stress concentration coefficient of the coal body was calculated and analysed, as shown in figure 6.

As shown in the figure, the maximum concentration coefficient of the abutment stress increases gradually with the caving height. Before the critical fracturing of KS1, the maximum stress concentration coefficient was 2.2 and 1.87 for KS2. Subsequently, with an increase in the fracturing height, it increased gradually. Before the fracturing of KS8, the concentration coefficient of the abutment stress was as high as 2.55. It is apparent that with the gradual upward development of the hard roof fracturing, the effect of hard roofs on the advanced coal body increases synchronously.

## 3.2. Impact characteristics for the first-time breaking of hard roofs

In the section above, the load on the hard roofs and its effect on the stress concentration of coal body in front of the working face are analysed. In this section, the impact characteristics of the breaking of each hard roof are studied. First, the maximum velocity and kinetic energy during the initial breaking and rotation of each hard roof are analysed.

### 3.2.1. Analysis of kinetic energy during first breaking

As shown in figure 7, the velocity distributed in each hard roof at the moment of collision with underlying caving rocks was obtained.

Figure 7a shows the velocity distribution at the moment of contact with the goaf gangue after KS1 was fractured and rotated. As shown, the velocity of the KS1 fractured block reached maximum at the middle, up to 18 m s$^{-1}$. Owing to the large mining thickness, the maximum vertical rotary subsidence of KS1 was as high as 17.55 m. Figure 7b shows the velocity distribution of the KS2 hard roof after breaking. As shown, the vertical rotary subsidence reached 16.6 m, and the velocity reached maximum at the middle. Before KS2 was in contact with the underlying caving strata, the underlying strata could maintain their integrity and were not affected by the KS2 fracturing impact.

As the caving height increased gradually, the rotation space beneath the hard roof became smaller owing to the hulking form of the caving strata; therefore, the maximum velocity of the breaking block reduced further. Figure 7c shows the velocity after KS4 had broken; the vertical rotary subsidence reached 11.8 m. With the further increase in the vertical distance between the hard roof and coal seam, the rotary space decreased accordingly. The vertical rotary subsidence of KS6 was 9.8 m, and the maximum velocity decreased as well, as shown in figure 7d.

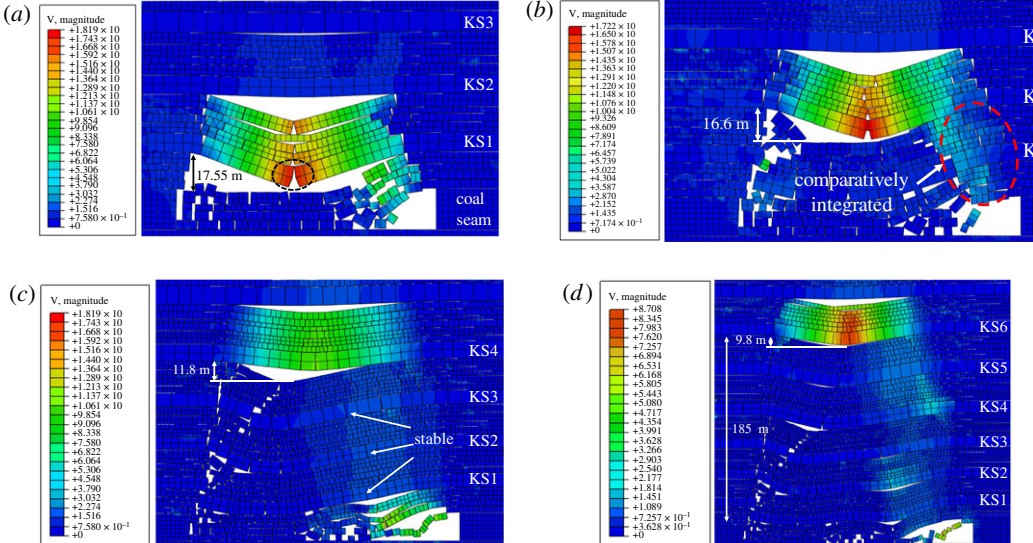

**Figure 7.** Speed characteristics during breakage of overlying strata. Breakage of (*a*) KS1, (*b*) KS2, (*c*) KS4 and (*d*) KS6.

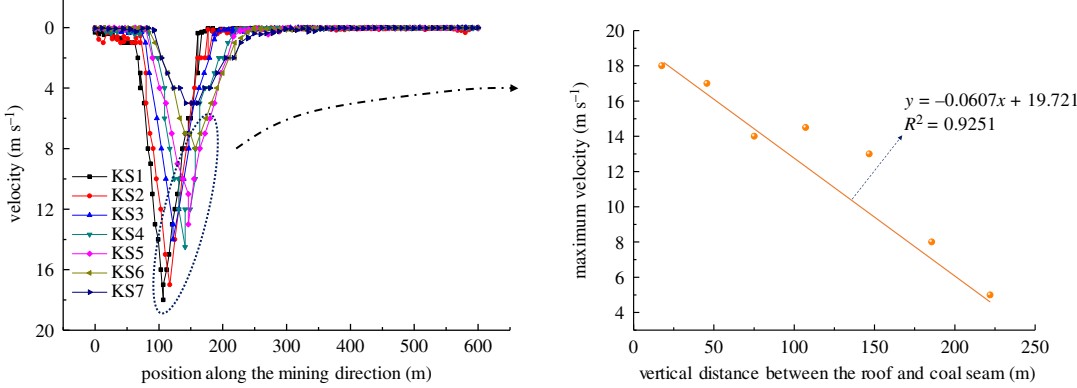

**Figure 8.** Speed comparison of overlying strata breaking.

For a comparative analysis of the velocity distribution after hard roof fracturing at different layers, the velocity distribution curve and variation law of the maximum velocity were obtained statistically, as shown in figure 8. The velocity distribution of the hard roof after fracturing presented a 'V' shape, and the velocity distributed in the hard rock was large in the middle and small at both sides. By comparing the maximum velocity of each hard roof after fracturing, it was found that the maximum velocity decreased linearly with the increase in the vertical distance between the hard roof and the coal seam.

Meanwhile, according to the simulation results, the breaking step of each hard roof from KS1 to KS7 is 96, 112, 113, 136, 147, 155 and 156 m, respectively. The unit weight of the rock layer is 25 kN m$^{-3}$, and the maximum kinetic energy of each rock layer after fracturing and rotation is as shown in figure 9.

With the fracturing and rotating of the hard rock, the potential energy of the hard rock and its overburden rock is transformed into kinetic energy. In the contacting moment between the hard rock and the underlying caving rock, the kinetic energy reached maximum. The maximum kinetic energy of the hard rock structure after rotation is related to the breaking step of the hard rock, the weight of the overburden rock and the rotation space. From the above analysis, it is apparent that with the caving height increasing upwards, the rotation space of each hard roof decreased, but the hard roof in the higher layer exhibited a large thickness and high strength, and its breaking step increased accordingly. Therefore, the distribution of the kinetic energy after hard roof fracturing does not necessarily present a monotonically increasing or decreasing distribution. As shown in figure 9, the kinetic energy of the KS4 structure is the highest, followed by that of KS5, reaching 7p × 10$^6$ J and 6 × 10$^6$ J, respectively.

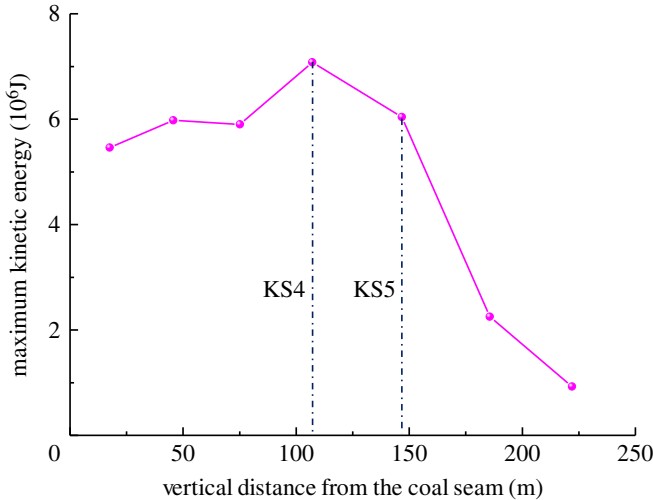

**Figure 9.** Maximum kinetic energy during strata breaking.

### 3.2.2. Analysis of first-breaking impact of hard roof

To further analyse the impact action of each hard roof, as shown in figure 10, the velocity distribution and structure characteristics of the underlying rock strata caused by the impact of the initial breaking of KS1–KS7 were obtained.

As shown in figure 10*a*, after the impact of KS1, part of the potential energy of the KS1 structure acted on the goaf gangue, resulting in the compaction of a goaf caving rock, while the other part resulted in the failure and instability of the underlying strata structure, which rotated synchronously with the KS1 hard rock structure, causing a certain impact on the working face. As shown in the figure, after the impact of the KS1 structure, the average velocity in the immediate roof reached 7 m s$^{-1}$, and may cause a large impact on the supports in the working face. Therefore, reasonable measures should be conducted to reduce the initial breaking step of the hard roof at the low level and weaken its action strength of ground pressure.

As shown in figure 10*b*, the impact of the KS2 structure induces fracturing and destabilizing of the KS1 structure stabilized below; furthermore, the KS2 structure, together with the KS1 structure, will rotate and move simultaneously. Owing to the limited breaking step and thickness of KS2, only part of its potential energy acted on the underlying rock strata, resulting in the fracturing and movement of KS1; however, the velocities of KS1 and the immediate roof were not large, being 3.0 m s$^{-1}$ and 3.6 m s$^{-1}$, respectively.

Figure 10*c* shows the velocity distribution and structural characteristics after the impact of KS3. Similarly, KS3 moves synchronously with KS1 and KS2, and the velocities at KS2, KS1 and the immediate roof are 2.8, 2.8 and 3.0 m s$^{-1}$, respectively. Owing to the large thickness and breaking span of KS4, the impact action of the KS4 rotary caused the underlying multilayer rock strata to break and move simultaneously, which easily caused a strong ground pressure in the stope. As shown in figure 10*d*, at this time, the velocities of all underlying hard strata and immediate roof under KS4 were relatively high, of 3.67, 3.7, 4.57 and 3.44 m s$^{-1}$, respectively.

As the level of the hard roof increases, the maximum velocity and potential energy decrease. With the underlying strata structures increasing further, it is more difficult for the high-level hard roof to cause synchronous breaking and rotatory of the underlying strata. The simulation study indicates that after the fracturing of KS5, KS6 and KS7, the underlying strata structures continue to maintain a certain stability, the velocity transmission from the high-level hard roof is weak, and the velocity distribution of each layer is different with a significant fluctuation, as shown in figure 10*e,f*.

Based on the analysis above, the velocity at the immediate roof caused by the impact of hard roofs at different levels are compared and analysed statistically, as shown in figure 11. The velocity at the immediate roof was the highest after the impact of KS1, reaching 7 m s$^{-1}$. This is primarily because the rotatory space and velocity were the highest after KS1 broke and rotated. In addition, no other stable structures appear under KS1, and the energy consumption was minimum; however, most of the energy acted on the gob rock. As the distance between the hard roof and coal seam increased, the underlying strata structure increased, and the energy consumed by those underlying structures

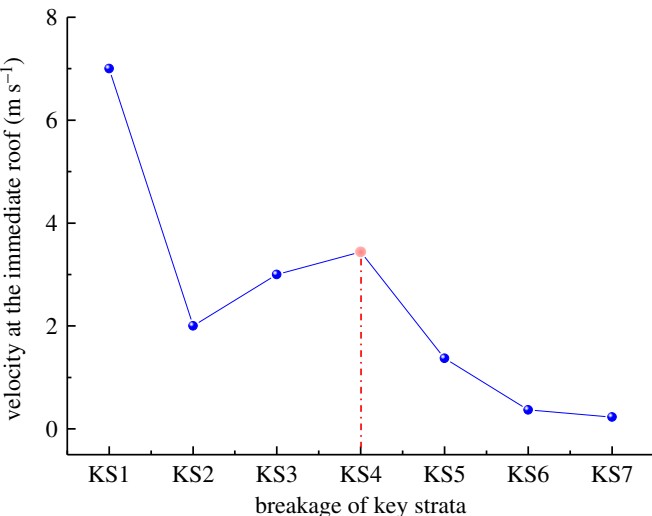

**Figure 10.** Dynamic impact characteristics of each key layer breaking. Dynamic characteristics during (*a*) KS1 impact, (*b*) KS2 impact, (*c*) KS3 impact, (*d*) KS4 impact, (*e*) KS5 impact and (*f*) KS6 impact.

**Figure 11.** Velocity of immediate roof after overlying strata impact action.

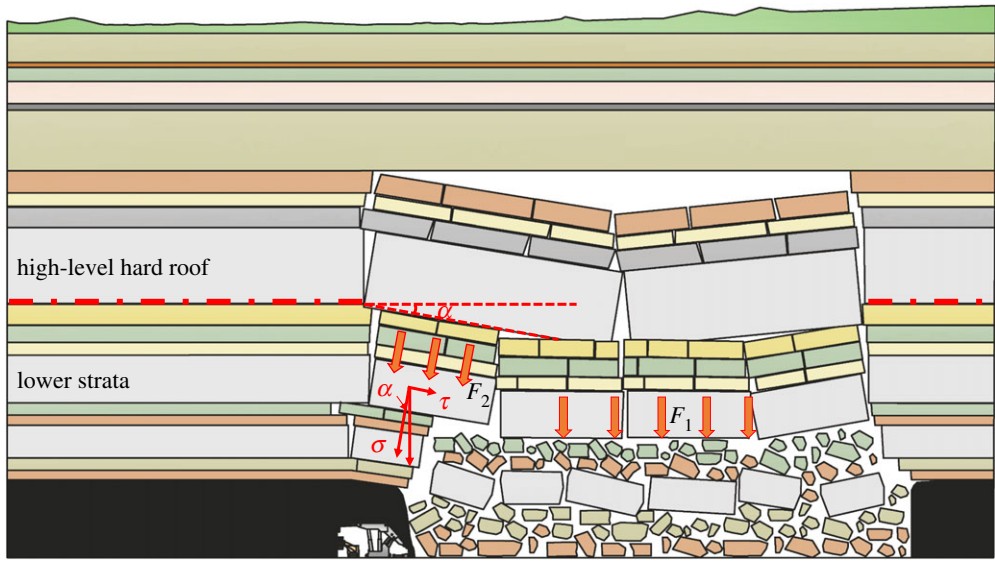

**Figure 12.** Ground pressure of far-field key layer breaking.

increased simultaneously. The instability and movement of the hard roofs resulted in the increase and then decrease of the velocity at the immediate roof. Furthermore, the breakage of KS4 resulted in the strongest impact on the working face.

The above analysis shows that a large space exists for the low-level hard roof to rotate, and most of the potential energy of the hard rock structure after breaking acts on the rock body in the gob. As the caving height develops upward, the rotation space decreases. Most of the potential energy of the hard roofs act on the instability and rotation of the underlying rock strata; thus, affecting the working face. However, as the vertical distance between the hard roof and coal seam increases, the effect of the hard roofs on the working face tends to decrease. According to the impact characteristics of each hard roof mentioned above, the impacts of the far-field KS4 and KS5 fracturing on the working face were the strongest.

## 4. Discussion

According to the numerical simulation results above, as the working face continued to be mined, the caving height developed layer by layer. Affected by the hulking form of the caving rock, the underlying separation space before the high-level hard roof breaking was smaller, and resulted in a decreased impact velocity. However, with a large thickness, high strength, and large breaking span, the breakage and rotation of the high-level hard roof would result in a large potential energy acting on the underlying strata; thus, causing the synchronous rotary movement of the lower strata. Furthermore, the strata synchronous movement in the large space would significantly impact the working face and surrounding coal and rock mass; thus, causing a strong ground pressure, as shown in figure 12.

With the increase in the vertical distance between the hard roof and coal seam, the separation space in the overburden will be further reduced, and the energy transferred to the coal seam will be further reduced after the breakage of the high-level hard roof; the hard roof fracture does not necessarily cause the instability and synchronous movement of the underlying strata. Considering the velocity transfer characteristics of hard roofs after fracturing and their effects on the coal and rock mass of the working face, in this geological condition, the hard roofs in the far field with the most serious effect on the working face were of KS4, followed by KS5. The vertical distances of KS4 and KS5 from the coal seam were 107 m and 146 m, respectively, and the ratios with the coal seam thickness were 5.35 and 7.3, respectively.

To further verify the simulation results, the hard roof fracture behaviours at different levels were monitored by a field measurement [3,27]. The thickness of the coal seam was 19 m, and the method of top-coal caving mining was adopted. The strata movement measurement points were arranged in the KS at different levels (22, 51 and 104 m from the coal seam). The thicknesses of the three KS were 12, 9.8 and 23 m from the bottom up. Meanwhile, the resistance characteristics of the working face support were recorded in real time. The monitoring results indicated that the support resistance in the

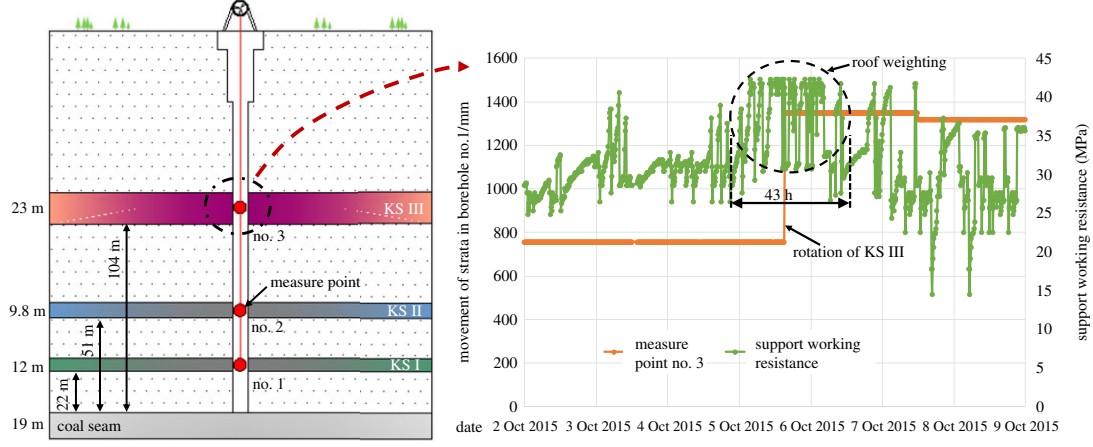

**Figure 13.** Ground pressure during the breaking of strata at different levels.

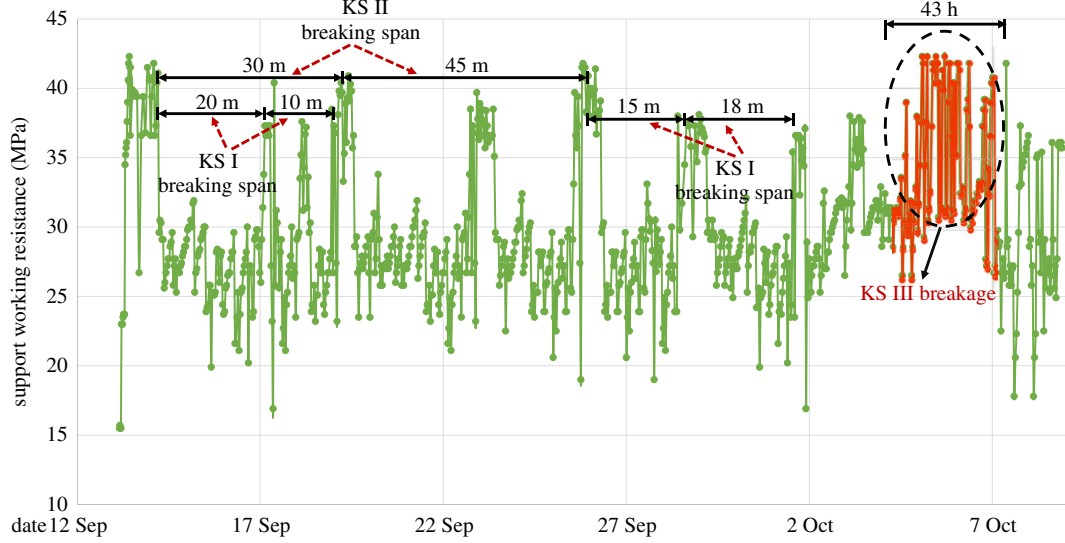

**Figure 14.** Complex ground pressure induced by the breaking of strata at different levels.

working face increased with the fracturing of the two KS, which were 22 and 51 m from the coal seam at the low level, and the dynamic load coefficients of the support were 1.15 and 1.34. The pressure durations were 7 and 16 h, and the working face had no apparent indications of a strong ground pressure. When the 23 m-thick key layer (which was 104 m from the coal seam) broke, the nos 35–95 supports in the working face were crushed, and the dynamic load coefficient of the support reached 1.54. The pressure duration reached 43 h, and the ratio of the distance between the highest key layer and the coal seam, to the coal seam thickness, was 5.47, as shown in figure 13.

To further analyse the weighting characteristics in the working face of different hard roof breakages, the whole pressure curve of the support in the middle of working face was counted, as shown in figure 14. Here, the pressure appeared many times in the support with the coal seam mining, the weighting steps of the KS I breakage ranged from 15 m to 25 m, during the weighting steps of KS II breakage ranged from 30 m to 45 m. Moreover, the support resistance during KS II breakage was greater than that during KS I breakage. When KS III was broken, the strength and duration of the pressure occurrence were much greater than those of KS I and KS II, which was the main factor causing the strong ground pressure. The alternate breaking of KS I, KS II and KS III resulted in the complex ground pressure on the working face.

As shown in the mining of the extra-thick coal seam with hard roofs, the large mining thickness of the coal seam resulted in a wide range of overburden migration. The foregoing research shows that owing to the large breaking span of high-level hard roofs, their failure and instability together with the synchronous fracture and rotation of the underlying strata, are the primary factors causing the strong

ground pressure on the working face. Based on the above studies, it was found that the thick and hard roofs with the ratio of the distance from the coal seam to the thickness of the coal seam between 5.3 and 7.3 exhibit the most serious impact on the working face.

## 5. Conclusion

(i) Herein, the numerical simulation software ABAQUS and the Mohr Coulomb criterion were used to establish a simulation model including all overlying strata. A cohesive unit with zero thickness was inserted into the solid unit of the same strata; moreover, the VUSDFLD subroutine was used to simulate the fracturing and caving of hard roofs; thus realizing the combination of finite element and discrete element.

(ii) The numerical calculation indicated that with the increase in the distance between the hard roof and the coal seam, the range of the bearing stress distributing on each hard roof increased and affected the stress distributed in the advanced coal body during the critical fracturing of the hard roofs. With the caving height developed upwards, the maximum concentration coefficient in the advanced coal body exhibited an increasing trend. It was apparent that the effect of the hard roofs overhanging on the advanced coal body increased with the increased occurrence levels of the hard roofs.

(iii) With the increase in the vertical distance between the hard roof and the coal seam, the separation space decreased, and the maximum velocity of the hard roof in the rotary process exhibited a linearly decreasing trend; however, the kinetic energy did not exhibit a monotonically increasing or decreasing trend, and the kinetic energies of the high-level hard roofs KS4 and KS5 were the highest. The breakage and rotatory of the hard roofs impacted the underlying strata; thus, resulting in the synchronous movement of the underlying strata. The kinetic energy of the hard strata at the low level acted primarily on the rock mass in the gob. Furthermore, the impact of the hard roof KS4 at the high level on the underlying strata was the strongest.

(iv) Combining the numerical calculation and field measurement, it was found that the breaking behaviour and impact action of the hard roof at different layers were distinct. The hard roof in the high level exhibited a large breaking span, and its failure and instability, together with the underlying rock strata, caused the strongest impact on the working face. Based on the studies above, it was found that the thick and hard roof with the ratio of the distance from the coal seam to the coal seam thickness between 5.3 and 7.3 demonstrated the most significant impact on the working face.

Data accessibility. The data of the figures in the text are provided as the electronic supplementary material accompanying this paper.

Authors' contributions. R.G. designed the study, participated in data analysis, carried out sequence alignments; B.H. participated in the design of the study and drafted the manuscript; H.X. carried out the numerical simulation and analysed the data; X.M. helped draft the manuscript and collected the field measurement data. All authors gave final approval for publication.

Competing interests. We declare we have no competing interests.

Funding. This work was supported by the State Key Research Development Program of China (grant no. 2018YFC0604500), and China Postdoctoral Science Foundation (grant no. 2019M651080), applied basic research project of Shanxi Province (grant no. 201901D211030), and Scientific and Technological Innovation Programs of Higher Education Institutions in Shanxi (STIP) (grant no. 2019L0208). The authors gratefully acknowledge the financial support from the organization.

Acknowledgements. We are very grateful to the editors and reviewers for their positive and constructive comments and suggestions. We are also grateful to Dr Yang Tai for his support in the numerical simulation, Dr Yiwen Lan and Tiejun Kuang's support in the field measurement, and also thank Editage (www.editage.cn) for English language editing.

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
