## [Reviewer comments · Royal Society Open Science]

Review History

RSOS-191383.R0 (Original submission)

Review form: Reviewer 1

Is the manuscript scientifically sound in its present form?

Yes

Are the interpretations and conclusions justified by the results?

Yes

Is the language acceptable?

Yes

Do you have any ethical concerns with this paper?

No

Have you any concerns about statistical analyses in this paper?

No

Recommendation?

Accept with minor revision (please list in comments)

Comments to the Author(s)

Review comments on the manuscript entitled 'Numerical simulation on fracturing behavior of hard roofs at different levels during extra-thick coal seam mining'

The comprehensive understanding of fracturing behavior and mechanism of hard roofs is of good interest to extra-thick coal seam mining. Facing with the different behavior of hard roofs at different levels, the numerical simulation was conducted to analyze the bearing stress distribution, fracturing morphology, and impact energy characteristics of hard roofs. Generally, the manuscript is of good quality, and well organized. Some problems are listed below and should be further addressed.

- 1) The main innovations of the manuscript need to be further clarified. Pure numerical simulation is not of great significance to comprehensive understanding of the mechanism and engineering benefits.
- 2) What's the main work you have done on the secondary development in the numerical simulation? The corresponding parts should be highlighted if they do exist.
- 3) The numerical simulation work is good, but how does the work be used to guide the mining practice?
- 4) Combining the numerical simulation and field measurement is a very good way to verify the correctness of numerical analysis results. A further discussion on this topic should be enhanced, but not only a short description.
- 5) According to the simulation results, KS4 has the strongest impact on the working face. Does it can be verified by the field measurement or other researches? Is the impacts limited in the stress distribution or other key issues? Does the simulation result only depend on the model or is it of universal significance?

Review form: Reviewer 2 (P. M. V. Nguyen)

Is the manuscript scientifically sound in its present form?

Yes

Are the interpretations and conclusions justified by the results?

Yes

Is the language acceptable?

Yes

Do you have any ethical concerns with this paper?

No

Have you any concerns about statistical analyses in this paper?

No

Recommendation?

Accept with minor revision (please list in comments)

Comments to the Author(s)

Although this manuscript is well-structured and written, there are some problems that need to be explained before publishing:

- In Chapter I, authors mentioned the disadvantages of UDEC I FLAC for modelling the fracturing effect of rock mass. Some references are given and they are all correct. However, there is no references that proved that ABAQUS is the right code for the case
- Please add a description of case study in chapter II?
- Where does the data in Table 1 come from? any reference?
- How have the horizontal stresses been measured in the field? what method? How to verify the horizontal stress was 1.2 times to the vertical stress?
- What is the set of data input for numerical modelling? The data in table 1 is not sufficient for modelling
- Please describe the modelling process
- What is the reference for figure 2
- Has the longwall support capacity been modelled?

Review form: Reviewer 3

Is the manuscript scientifically sound in its present form?

Yes

Are the interpretations and conclusions justified by the results?

Yes

Is the language acceptable?

Yes

Do you have any ethical concerns with this paper?

No

Have you any concerns about statistical analyses in this paper?

No

Recommendation?

Accept with minor revision (please list in comments)

Comments to the Author(s)

Manuscript Number: RSOS-191383

Full Title: Numerical simulation on fracturing behavior of hard roofs at different levels during extra-thick coal seam mining

Comments:

The manuscript presents the fracturing behavior of hard roofs at different levels during extra-thick coal seam mining. The results reveal the mechanism of strong ground pressure during coal seam mining with multiple hard roofs, which have a practical significance to determine the control target of the hard roofs. The research is very meaningful, and the manuscript is well written. I suggest the manuscript to be accepted after some minor revisions. The specific comments and suggestions are shown as follows.

- (1) Figure 3, why are the curves so deviation? How to determine the changing of these curves?
- (2) Line 32, Page 7, the presentation of "KS1" should be "KS2".
- (3) Figure 5, the stress concentration coefficient increases gradually with the caving height, but why the stress concentration coefficient during KS1 breaking is larger than KS2 ?
- (4) Figure 7, there are two coordinate systems in one figure which is not common.

- (5) Under the impact of each hard roof, as shown in figure 9, what does the velocity distribution of the underlying strata reflect?
- (6) Figure 8, how to calculate the kinetic energy during the hard roof breaking and rotating?
- (7) Line 19, Page 2, the sentence "However, this behavior in different levels are different, and the characteristics, morphology, and impact effect of hard roofs in different levels are unclear." is very hard to read.
- (8) There are a number of grammatical errors and badly constructed sentences which bring confusion. Please go through the paper carefully to improve the English writing.

Decision letter (RSOS-191383.R0)

24-Sep-2019

Dear Dr Gao,

The editors assigned to your paper ("Numerical simulation on fracturing behavior of hard roofs at different levels during extra-thick coal seam mining") have now received comments from reviewers. We would like you to revise your paper in accordance with the referee and Associate Editor suggestions which can be found below (not including confidential reports to the Editor). Please note this decision does not guarantee eventual acceptance.

Please submit a copy of your revised paper before 17-Oct-2019. Please note that the revision deadline will expire at 00.00am on this date. If we do not hear from you within this time then it will be assumed that the paper has been withdrawn. In exceptional circumstances, extensions may be possible if agreed with the Editorial Office in advance. We do not allow multiple rounds of revision so we urge you to make every effort to fully address all of the comments at this stage. If deemed necessary by the Editors, your manuscript will be sent back to one or more of the original reviewers for assessment. If the original reviewers are not available, we may invite new reviewers.

- Data accessibility

<http://datadryad.org/submit?journalID=RSOS&manu=RSOS-191383>

- Competing interests

- Authors' contributions

- Acknowledgements

- Funding statement

Kind regards,

Andrew Dunn

on behalf of Prof R. Kerry Rowe (Subject Editor)

Associate Editor's comments:

The reviewers consider your paper to be a reasonable contribution, though they each point out a number of ways the manuscript may be improved. Please ensure you incorporate the required changes and provide a point-by-point response to these queries in your revision.

Comments to Author:

Reviewers' Comments to Author:

Reviewer: 1

Comments to the Author(s)

Review comments on the manuscript entitled 'Numerical simulation on fracturing behavior of hard roofs at different levels during extra-thick coal seam mining'

The comprehensive understanding of fracturing behavior and mechanism of hard roofs is of good interest to extra-thick coal seam mining. Facing with the different behavior of hard roofs at different levels, the numerical simulation was conducted to analyze the bearing stress distribution, fracturing morphology, and impact energy characteristics of hard roofs. Generally, the manuscript is of good quality, and well organized. Some problems are listed below and should be further addressed.

- 1) The main innovations of the manuscript need to be further clarified. Pure numerical simulation is not of great significance to comprehensive understanding of the mechanism and engineering benefits.
- 2) What's the main work you have done on the secondary development in the numerical simulation? The corresponding parts should be highlighted if they do exist.
- 3) The numerical simulation work is good, but how does the work be used to guide the mining practice?
- 4) Combining the numerical simulation and field measurement is a very good way to verify the correctness of numerical analysis results. A further discussion on this topic should be enhanced, but not only a short description.
- 5) According to the simulation results, KS4 has the strongest impact on the working face. Does it can be verified by the field measurement or other researches? Is the impacts limited in the stress distribution or other key issues? Does the simulation result only depend on the model or is it of universal significance?

Reviewer: 2

Comments to the Author(s)

Although this manuscript is well-structured and written, there are some problems that need to be explained before publishing:

- In Chapter I, authors mentioned the disadvantages of UDEC I FLAC for modelling the fracturing effect of rock mass. Some references are given and they are all correct. However, there is no references that proved that ABAQUS is the right code for the case
- Please add a description of case study in chapter II?
- Where does the data in Table 1 come from? any reference?
- How have the horizontal stresses been measured in the field? what method? How to verify the horizontal stress was 1.2 times to the vertical stress?

- What is the set of data input for numerical modelling? The data in table 1 is not sufficient for modelling
- Please describe the modelling process
- What is the reference for figure 2
- Has the longwall support capacity been modelled?

Reviewer: 3

Comments to the Author(s)

Manuscript Number: RSOS-191383

Full Title: Numerical simulation on fracturing behavior of hard roofs at different levels during extra-thick coal seam mining

Comments:

The manuscript presents the fracturing behavior of hard roofs at different levels during extra-thick coal seam mining. The results reveal the mechanism of strong ground pressure during coal seam mining with multiple hard roofs, which have a practical significance to determine the control target of the hard roofs. The research is very meaningful, and the manuscript is well written. I suggest the manuscript to be accepted after some minor revisions. The specific comments and suggestions are shown as follows.

- (1) Figure 3, why are the curves so deviation? How to determine the changing of these curves?
- (2) Line 32, Page 7, the presentation of "KS1" should be "KS2".
- (3) Figure 5, the stress concentration coefficient increases gradually with the caving height, but why the stress concentration coefficient during KS1 breaking is larger than KS2 ?
- (4) Figure 7, there are two coordinate systems in one figure which is not common.
- (5) Under the impact of each hard roof, as shown in figure 9, what does the velocity distribution of the underlying strata reflect?
- (6) Figure 8, how to calculate the kinetic energy during the hard roof breaking and rotating?
- (7) Line 19, Page 2, the sentence "However, this behavior in different levels are different, and the characteristics, morphology, and impact effect of hard roofs in different levels are unclear." is very hard to read.
- (8) There are a number of grammatical errors and badly constructed sentences which bring confusion. Please go through the paper carefully to improve the English writing.

Author's Response to Decision Letter for (RSOS-191383.R0)

See Appendix A.

RSOS-191383.R1 (Revision)

Review form: Reviewer 1

Is the manuscript scientifically sound in its present form?

Yes

Are the interpretations and conclusions justified by the results?

Yes

Is the language acceptable?

No

Do you have any ethical concerns with this paper?

No

Have you any concerns about statistical analyses in this paper?

No

Recommendation?

Accept with minor revision (please list in comments)

Comments to the Author(s)

- 1) The main contribution of the secondary development in the numerical simulation should be explained in the manuscript.
- 2) The English should be further carefully checked and revised. For example, on Page 24, Line 14, what does the expression 'hulking sex of the caving rock' mean?

Review form: Reviewer 2 (P. M. V. Nguyen)

Is the manuscript scientifically sound in its present form?

Yes

Are the interpretations and conclusions justified by the results?

Yes

Is the language acceptable?

Yes

Do you have any ethical concerns with this paper?

No

Have you any concerns about statistical analyses in this paper?

No

Recommendation?

Accept as is

Comments to the Author(s)

Your response to my comments and suggestions was clear. Thank you!

Decision letter (RSOS-191383.R1)

30-Oct-2019

Dear Dr Gao:

Manuscript ID RSOS-191383.R1 entitled "Numerical simulation on fracturing behavior of hard roofs at different levels during extra-thick coal seam mining" which you submitted to Royal

Society Open Science, has been reviewed. The comments of the reviewer(s) are included at the bottom of this letter.

Please submit a copy of your revised paper before 22-Nov-2019. Please note that the revision deadline will expire at 00.00am on this date. If we do not hear from you within this time then it will be assumed that the paper has been withdrawn. In exceptional circumstances, extensions may be possible if agreed with the Editorial Office in advance. We do not allow multiple rounds of revision so we urge you to make every effort to fully address all of the comments at this stage. If deemed necessary by the Editors, your manuscript will be sent back to one or more of the original reviewers for assessment. If the original reviewers are not available we may invite new reviewers.

- Ethics statement

- Data accessibility

- Competing interests

- Authors' contributions

- Acknowledgements

- Funding statement

Kind regards,
Anita Kristiansen
Editorial Coordinator
Royal Society Open Science
openscience@royalsociety.org

on behalf of R. Kerry Rowe (Subject Editor)
openscience@royalsociety.org

Associate Editor Comments to Author:

Comments to the Author:

While the reviewers consider the scientific quality of your work to be suitable, Reviewer 1 has observed that the quality of the written English is below the standard that we would expect.

Please use a service listed at <https://royalsociety.org/journals/authors/language-polishing/> to seek language editing support. You must not only include a fully revised manuscript in your revision, but also provide evidence that you've had your manuscript checked by a language editing service. Failure to do so may result in your paper being rejected.

Reviewer comments to Author:

Reviewer: 1

Comments to the Author(s)

1) The main contribution of the secondary development in the numerical simulation should be explained in the manuscript.

2) The English should be further carefully checked and revised. For example, on Page 24, Line 14, what does the expression 'hulking sex of the caving rock' mean?

Reviewer: 2

Comments to the Author(s)

Your response to my comments and suggestions was clear. Thank you!

Author's Response to Decision Letter for (RSOS-191383.R1)

See Appendix B.

RSOS-191383.R2 (Revision)

Review form: Reviewer 1

Is the manuscript scientifically sound in its present form?

Yes

Are the interpretations and conclusions justified by the results?

Yes

Is the language acceptable?

Yes

Do you have any ethical concerns with this paper?

No

Have you any concerns about statistical analyses in this paper?

No

Recommendation?

Accept as is

Comments to the Author(s)

WELL DONE. NO FURTHER QUESTIONS.

Decision letter (RSOS-191383.R2)

09-Dec-2019

Dear Dr Gao,

It is a pleasure to accept your manuscript entitled "Numerical simulation on fracturing behavior of hard roofs at different levels during extra-thick coal seam mining" in its current form for publication in Royal Society Open Science. The comments of the reviewer(s) who reviewed your manuscript are included at the foot of this letter.

Kind regards,
Anita Kristiansen
Editorial Coordinator
Royal Society Open Science
openscience@royalsociety.org

on behalf of R. Kerry Rowe (Subject Editor)
openscience@royalsociety.org

Associate Editor Comments to Author:
Comments to the Author:
The reviewer is happy with the revision.

Reviewer comments to Author:
Reviewer: 1

Comments to the Author(s)
WELL DONE. NO FURTHER QUESTIONS.

Appendix A

Dear editors and reviewers,

On behalf of my co-authors, we thank you very much for giving us an opportunity to revise our manuscript. We are grateful to the editor and reviewers for their positive and constructive comments and suggestions on our manuscript. These comments have all been of great help to us in the revision and improvement of our paper, as well as providing significant guidance for our research.

We have revised our manuscript according to the comments from the reviewers. The main corrections to the paper and our responses to the reviewers are as follows:

Associate Editor's comments:

The reviewers consider your paper to be a reasonable contribution, though they each point out a number of ways the manuscript may be improved. Please ensure you incorporate the required changes and provide a point-by-point response to these queries in your revision.

Comments to Author:

Reviewers' Comments to Author:

Reviewer: 1

Comments to the Author(s)

Review comments on the manuscript entitled 'Numerical simulation on fracturing behavior of hard roofs at different levels during extra-thick coal seam mining'

The comprehensive understanding of fracturing behavior and mechanism of hard roofs is of good interest to extra-thick coal seam mining. Facing with the different behavior of hard roofs at different levels, the numerical simulation was conducted to analyze the bearing stress distribution, fracturing morphology, and impact energy characteristics of hard roofs. Generally, the manuscript is of good quality, and well organized. Some problems are listed below and should be further addressed.

1) The main innovations of the manuscript need to be further clarified. Pure numerical simulation is not of great significance to comprehensive understanding of the mechanism and engineering benefits.

Reply: Thanks for your comments. The methods of numerical simulation and in-situ investigation were used to reveal the ground pressure mechanism induced by the hard roofs. There was not much explanation about the field measurement for we have published the in-situ

measurement individually, as could be found in the paper of ‘*Yiwen Lan, Rui Gao, Bin Yu et al. In situ studies on the characteristics of strata structures and strata behaviors hazards in mining a super-thick coal seam with hard roof. Energies, 2018, 11(9), 2470.*’ For the richness and completeness of the article, we added an in-situ measurement conceptual figure and in-situ measurement results, also, the published paper was also been cited. As can be found in the “Discussion section” in the revised manuscript.

2) What’s the main work you have done on the secondary development in the numerical simulation? The corresponding parts should be highlighted if they do exist.

Reply: Thanks for your comments. In the simulation, a cohesive unit with zero thickness was inserted globally into the solid element of the same rock stratum, and a VUSDFLD subroutine was utilized to simulate the fracturing and caving of rock stratum.

3) The numerical simulation work is good, but how does the work be used to guide the mining practice?

Reply: Thanks for your comments. By the numerical simulation, the effects of each hard roof breakage on the advanced stress distribution and coal body failure were gained, thus, the complex ground pressure mechanism was revealed. Through the simulation results, we could know whether the strength of ground pressure caused by the breaking of hard roofs in different layers has a direct impact on the mining production. More importantly, the research results are of great significance for guiding the selection of the target layer of hard roof control. The research results have a guiding effect on coal mining under similar geological conditions.

4) Combining the numerical simulation and field measurement is a very good way to verify the correctness of numerical analysis results. A further discussion on this topic should be enhanced, but not only a short description.

Reply: Thanks for your comments. For the completeness and richness of description, we add an in-situ measurement conceptual figure and a measurement pressure curve in the ‘discussion section’, in addition, more description has also been added, as marked in red color in ‘discussion section’.

5) According to the simulation results, KS4 has the strongest impact on the working face. Does it can be verified by the field measurement or other researches? Is the impacts limited in the stress

distribution or other key issues? Does the simulation result only depend on the model or is it of universal significance?

Reply: Thanks for your comments. In the simulation study, the KS4 has the strongest impact on the working face, the ratio between the distance of the KS4 from the coal seam to the coal seam thickness was 5.35. In the in-situ measurement study, the KSIII breakage is the main reason causing the strong ground pressure, the ratio between the distance of the KSIII from the coal seam to the coal seam thickness was 5.47, which was similar to that of KS4. The simulation and in-situ results could be verified mutually. The high-level thick hard roof breakage mainly impacts the surrounding rock of the working face, which mainly shows stress concentration and surrounding rock failure.

The results of numerical simulation are mainly influenced by the establishment of the model, and the measured results and the simulation results correspond to each other which proved the conclusion to be right. The research conclusion are of significance in guiding the ground pressure of the hard roof mining area under similar conditions.

Once again, thank you very much for your comments and suggestions.

Reviewer: 2

Comments to the Author(s)

Although this manuscript is well-structured and written, there are some problems that need to be explained before publishing:

(1) In Chapter I, authors mentioned the disadvantages of UDEC I FLAC for modelling the fracturing effect of rock mass. Some references are given and they are all correct. However, there is no references that proved that ABAQUS is the right code for the case

Reply: Thanks for your comments. What you said is absolutely right. These references only prove that UDEC and FLAC cannot fully simulate the fracturing effect we want to achieve, also, they cannot prove that ABAQUS can be used to simulate. In the manuscript, we used ABAQUS for an exploration research, and the results showed that rock fracturing, impact strength and stress distribution can be reflected, proving that ABAQUS can be used to simulate and analyze rock fracturing and can achieve the effect we wanted. The simulation results were also been verified by the in-situ measurement.

(2) Please add a description of case study in chapter II?

Reply: Thanks for your comments. To better understand the mining situation of the study background, we added a picture as seen of Figure 1. The situation of mining working face and the overlying hard rock strata distribution were given.

(3) Where does the data in Table 1 come from? any reference?

Reply: Thanks for your comments. We are very sorry that we made some mistakes in Table 1, and we have corrected the data. The basic parameters of each layer were gained by field drilling and coring test. Furthermore, based on the in-situ measurement data and conversion, the simulation parameters were obtained. The steps are shown below and the description has been add in the section 2. (*Yang Tai, Xiaole Han, Peng Huang and Baifu An. The mining pressure in mixed workface using a gangue backfilling and caving method. Journal of Geophysics and Engineering (2019) 16, 1-15.*)

(4) How have the horizontal stresses been measured in the field? what method? How to verify the horizontal stress was 1.2 times to the vertical stress?

Reply: Thanks for your comments. The hydraulic fracturing was used to measure the in-situ

stress distribution in the rock formation. In the study, a manuscript written by a well-known academician was quoted (*Kang Hongpu, Yi Bingding, Gao Fuqiang, et al. Database and characteristics of underground in-situ stress distribution in Chinese coal mines[J]. Journal of China Coal Society, 2019, 44(1): 23-33*). In the manuscript, the relationship between the average horizontal stress and the vertical stress was gained.

$$K_{av} = \frac{129.58}{H} + 0.606 \quad (1)$$

Where K_{av} is the ratio of horizontal stress to vertical stress, H is the burial depth.

To add an average coefficient of lateral pressure in the model, five values of burial depth were selected, as 490m, 390m, 290m, 190m, 90m and an average value K_{av} of 1.2 was obtained. Thus, the value 1.2 of the ratio of horizontal stress to vertical stress was selected in the simulation.

(5) What is the set of data input for numerical modelling? The data in table 1 is not sufficient for modelling

Reply: Thanks for your comments. We are very sorry that we made some mistakes in table 1, and in the revised paper, the data in the table 1 have been corrected. The physical and mechanical parameters of each layer were given. In addition, to realize the effect of rock fracturing and caving, a cohesive unit with zero thickness was inserted globally into the solid element of the same rock stratum, and a VUSDFLD subroutine was utilized to simulate the fracturing and caving of rock stratum.

(6) Please describe the modelling process

Reply: Thanks for your comments. The modelling process mainly includes the following steps: (1) Determine the size of the numerical model. 492 m × 600 m is selected in the paper, a 50-m coal pillar was set, and the working face was mined continuously for a length of 500 m. (2) Give the parameters of coal seam and overlying strata. The basic parameters of each layer were gained by field drilling and coring test. Furthermore, based on the in-situ measurement data and conversion, the simulation parameters were obtained (*Reference No.24*). (3) Meshing and boundary condition. The mesh size was between 3 to 6 m, with a total of 14,000 grids. The horizontal displacements on both sides of the model and the horizontal and vertical displacements at the bottom were fixed. (4) Loading horizontal stress. The horizontal stress was 1.2 times to the vertical stress. (5) Constitutive model. The Mohr–Coulomb criterion was used in the model, and the bilinear constitutive model and the maximum nominal stress failure criterion were adopted in this project. (6) Rock caving. a cohesive

unit with zero thickness was inserted globally into the solid element of the same rock stratum, and a VUSDFLD subroutine was utilized to simulate the fracturing and caving of rock stratum.

(7) What is the reference for figure 2.

Reply: Thanks for your comments. We quoted the reference of “Rui Gao, Hongchun Xia, Kun Fang and Chunwang Zhang. Dome Roof Fall Geohazards of Full-Seam Chamber with Ultra-Large Section in Coal Mine. Applied Sciences. 2019, 9, 3891”. In the revised paper, we also added the reference in the list (No. 26).

(8) Has the longwall support capacity been modelled?

Reply: Thanks for your comments. We are sorry that we did not simulating the longwall support capacity as the fracturing and caving effect and the stress distribution in the rock mass were mainly been concerned in the paper. The structure features and impact strength of hard rock fracturing and the corresponding changes of overburden stress were the main study object. In our future research, according to your valuable advice, the change law of longwall support capacity would be studied.

Once again, thank you very much for your comments and suggestions. Your suggestions are very professional and of great significance to the improvement of the paper, we all appreciate it.

Reviewer: 3

Comments to the Author(s)

Manuscript Number: RSOS-191383

Full Title: Numerical simulation on fracturing behavior of hard roofs at different levels during extra-thick coal seam mining

Comments:

The manuscript presents the fracturing behavior of hard roofs at different levels during extra-thick coal seam mining. The results reveal the mechanism of strong ground pressure during coal seam mining with multiple hard roofs, which have a practical significance to determine the control target of the hard roofs. The research is very meaningful, and the manuscript is well written. I suggest the manuscript to be accepted after some minor revisions. The specific comments and suggestions are shown as follows.

(1) Figure 3, why are the curves so deviation? How to determine the changing of these curves?

Reply: Thanks for your comments. In the numerical simulation, the joint planes were inserted into the finite element unit to realize the fracturing of rock strata, thus, the stress distribution of rock strata presented certain dispersions and deviations; however, the law was still obvious, which could reflect the law of stress distribution of rock strata. To obtain the variation rule of these curves, we did a simple manipulation of the data, as seen in the following, but to make the simulation data truthful, we kept the original data.

Figure 3 stress distribution in KS1

(2) Line 32, Page 7, the presentation of “KS1” should be “KS2”.

Reply: Thanks for your comments. We have corrected the mistake.

(3) Figure 5, the stress concentration coefficient increases gradually with the caving height, but why the stress concentration coefficient during KS1 breaking is larger than KS2 ?

Reply: Thanks for your comments. The concentration coefficient in the advanced coal body is related to the location of the overlying key strata, the breaking span and the vertical distance from the coal seam, so it's variable. As seen from the Figure, the breaking span of KS1 and KS2 is approximate, but the KS1 is closer to the coal seam, and has a more influence on the advanced coal body.

(4) Figure 7, there are two coordinate systems in one figure which is not common.

Reply: Thanks for your comments. We have corrected the problem as seen of Figure 8 in the revised paper.

(5) Under the impact of each hard roof, as shown in figure 9, what does the velocity distribution of the underlying strata reflect?

Reply: Thanks for your comments. First of all, when the hard roof breaking down, the underlying layers were assigned a certain speed, the breakage of hard strata has a certain impact on the underlying strata. In addition, the speed of the underlying layers is different, the changing rule of the speed was not obvious, indicated an interaction between the various rock formations. Meanwhile, according to the speed of each layer, the impact strength of the hard rock breaking can also be determined.

(6) Figure 8, how to calculate the kinetic energy during the hard roof breaking and rotating?

Reply: Thanks for your comments. According to the theory of key strata, when the key layer broken, the overlying strata moved synchronously. Thus, based on the calculation formula of kinetic energy.

$$E_k = \frac{1}{2}mv^2 \quad (1)$$

Where the E_k is kinetic energy, m is common quality of key strata and its synchronous moved overlying strata, v could be obtained from Figure 8.

(7) Line 19, Page 2, the sentence “However, this behavior in different levels are different, and the characteristics, morphology, and impact effect of hard roofs in different levels are unclear.” is very hard to read.

Reply: Thanks for your comments. We have corrected the expression of the sentence, as seen to be “However, the characteristics, morphology, and impact effect of hard roofs in different levels are

different and unclear” in the revised paper.

(8) There are a number of grammatical errors and badly constructed sentences which bring confusion. Please go through the paper carefully to improve the English writing.

Reply: Thanks for your comments. We have corrected all the wrong and misleading sentences in the revised paper, as marked in the red color.

Once again, thank you very much for your comments and suggestions.

Appendix B

Dear editors and reviewers,

On behalf of my co-authors, we thank you very much for giving us an opportunity to revise our manuscript. We are grateful to the editor and reviewers for their positive and constructive comments and suggestions on our manuscript. These comments have all been of great help to us in the revision and improvement of our paper, as well as providing significant guidance for our research.

We have revised our manuscript according to the comments from the reviewers. The main corrections to the paper and our responses to the reviewers are as follows:

Comments to Author:

Reviewers' Comments to Author:

Reviewer: 1

Comments to the Author(s)

1) The main contribution of the secondary development in the numerical simulation should be explained in the manuscript.

Reply: Thanks for your comments. The secondary development in the numerical simulation was to realize the fracturing and caving of rock strata, and was operated in the following steps: firstly, the rand function was used to generate the corresponding slice in the model to simulate the cracks in the rock strata. Secondly, a cohesive unit with zero thickness was inserted at each slice. At last, define the constitutive model of the cohesive element. The constitutive model of the cohesive element was Mohr Coulomb, which was shown as Figure 3. A VUSDFLD subroutine was utilized to invoke the function; thus, realizing the fracturing and caving of the rock strata. The relevant expression have been added in the revised manuscript.

2) The English should be further carefully checked and revised. For example, on Page 24, Line 14, what does the expression 'hulking sex of the caving rock' mean?

Reply: Thanks for your comments. We have corrected the expression 'hulking sex of the caving rock' to 'hulking form of the caving rock', which means that the rock strata have the hulking features after caving. Furthermore, we have edited the English language of the manuscript by a professional

language polishing agency. The relevant corrections are shown in the revised paper which are marked in red color.

Reviewer: 2

Comments to the Author(s)

Your response to my comments and suggestions was clear. Thank you!

Reply: Thanks very much for your comments and suggestions in round 1, we all appreciate it.

Once again, thank you very much for your comments and suggestions.